# Recent Technical Progression in Photoacoustic Imaging—Towards Using Contrast Agents and Multimodal Techniques

Zuomin Zhao [1,*] and Teemu Myllylä [1,2]

1 Optoelectronics and Measurement Techniques Unit, Faculty of Information Technology and Electrical Engineering, University of Oulu, 90014 Oulu, Finland; teemu.myllyla@oulu.fi
2 Research Unit of Medical Imaging, Physics and Technology, Faculty of Information Technology and Electrical Engineering, University of Oulu, 90220 Oulu, Finland
* Correspondence: zuomin.zhao@oulu.fi

**Abstract:** For combining optical and ultrasonic imaging methodologies, photoacoustic imaging (PAI) is the most important and successful hybrid technique, which has greatly contributed to biomedical research and applications. Its theoretical background is based on the photoacoustic effect, whereby a modulated or pulsed light is emitted into tissue, which selectively absorbs the optical energy of the light at optical wavelengths. This energy produces a fast thermal expansion in the illuminated tissue, generating pressure waves (or photoacoustic waves) that can be detected by ultrasonic transducers. Research has shown that optical absorption spectroscopy offers high optical sensitivity and contrast for ingredient determination, for example, while ultrasound has demonstrated good spatial resolution in biomedical imaging. Photoacoustic imaging combines these advantages, i.e., high contrast through optical absorption and high spatial resolution due to the low scattering of ultrasound in tissue. In this review, we focus on advances made in PAI in the last five years and present categories and key devices used in PAI techniques. In particular, we highlight the continuously increasing imaging depth achieved by PAI, particularly when using exogenous reagents. Finally, we discuss the potential of combining PAI with other imaging techniques.

**Keywords:** photoacoustic; imaging modality; optical; ultrasonic; brain

## 1. Introduction

Ultrasonography and other related ultrasonic techniques are safe methods that are widely used in medical imaging, offering a deep imaging depth, high spatial resolution and a low device cost. However, ultrasonic contrast is low in different soft tissues, hampering the application of these techniques to functional and molecular tissue imaging. Optical imaging techniques can provide molecular information about tissues and organs based on their optical properties, although with a limited spatial accuracy and imaging depth. For instance, diffuse optical tomography (DOT) has made great advances in terms of its imaging of organs and the blood circulation in peripherals [1], but in deep tissue, at the depth of several centimeters, its spatial resolution is only 5–10 mm. In order to provide both high contrast and high spatial resolution in deep tissue imaging, some form of hybrid imaging technique is most likely to succeed, specifically one that combines acoustics and optics. Photoacoustic imaging (PAI) and acoustic-optical tomography (AOT) were introduced at the end of the last century. AOT is still in its infancy [2], but PAI has achieved considerable progress [3–5] in preclinical imaging, mainly due to the relatively low attenuation of PA signals in highly turbid tissues. Furthermore, by combining PAI with ultrasonography techniques, it is possible to gather both structural and functional information [6], while combining PAI with DOT allows a quantitative detection of biological chromophores in tissues [7].

### 1.1. PAI Technique and Its Categories

PAI is based on the photoacoustic (PA) effect, first reported by Alexander Graham Bell in 1880. The PA effect is produced by the absorption of modulated light (or electromagnetic radiation) in a material. The absorbed optical energy is converted into heat by non-radiative relaxation, resulting in a pressure variation. This fast pressure variation is released through acoustic waves, which propagate outside the material. By applying acoustic (usually ultrasonic) detectors to receive acoustic waves and scanning the PA generation zone, it is possible to obtain an optical absorbing image of the material through a specific image reconstruction algorithm. This technique is known as PA (or optoacoustic) imaging and has found wide application in biotissue studies and in vivo small animal studies [8,9]. Examples include visualizing blood vessels [10,11], imaging tissue tumours [12,13], determining blood hemoglobin (HbR) [14] and deoxyhemoglobin (HbO) [15] levels as well as oxygen saturation [16,17] and oxygen metabolism [15,18]. It is also used in measuring blood flow [15,19], counting melanomas and cancerous cells in blood [20,21] and sensing fatty deposits in blood vessels [22,23]. Together with exogenous contrast agents [24], such as bio-compatible nanoparticles, chemical dyes and report genes, researchers have greatly expanded the PAI application range, while improving imaging depth and quality in animal and human tissues alike.

Depending on their imaging resolution and depth, PAI methods are approximately divided into PA microscopy (PAM), PA macroscopy (PAMac) and PA computed tomography (PAT or PACT). PAM can be further classified as optical-resolution microscopy (OR-PAM) and acoustic-resolution microscopy (AR-PAM), while PAT can be classified as single, linear-array, ring-array, planar-array, cylindrical-array and spherical-array PAT, based on the detector number and shape [25]. All of these imaging techniques have their own properties and application areas. OR-PAM utilizes ballistic and quasi-ballistic photons to generate images and provides the highest resolution, to the micrometer or even sub-micrometer range. As shown in Figure 1a, the exciting light and detected ultrasound are co-focused. However, OR-PAM's imaging depth in brain tissue is less than 1 mm. AR-PAM and PAMac, on the other hand, use diffusive light to generate images with a lateral resolution from a few tens of micrometers to sub-millimeters, depending on the transducer frequency and focus. The imaging depth of AR-PAM and PAMac in brain tissue can be up to 3–5 mm and 10–20 mm, respectively, at red and near-infrared wavelengths. Due to its limited imaging depth, PAM is mainly suited to small animal imaging, for which it has achieved great success in brain cortex studies. Though similar to PAM, PAMac has a greater imaging depth, achieved by sacrificing lateral resolution to some degree, which is a result of using transducers with a lower frequency and a smaller acoustic focus (Figure 1b). Differing from both PAM and PAMac, in which the detector receives PA waves along or nearly along its acoustic axis, PACT usually receives PA waves from the imaging plane perpendicular to the direction of excitation light. As a result, it has the capacity to image tissues up to a depth of several centimeters with sub-millimeter resolution. Hence, while PAT is capable of imaging the entire body or head of a small animal, it can also produce images of the outer layers of the human brain, which is covered by a scalp and skull with a thickness of up to 10 mm. Although single-detector PAT has a simpler setup, array-detector PAT offers faster data acquisition without scanning or with a shorter scanning period, making it more suitable for real-time imaging (Figure 1c,d).

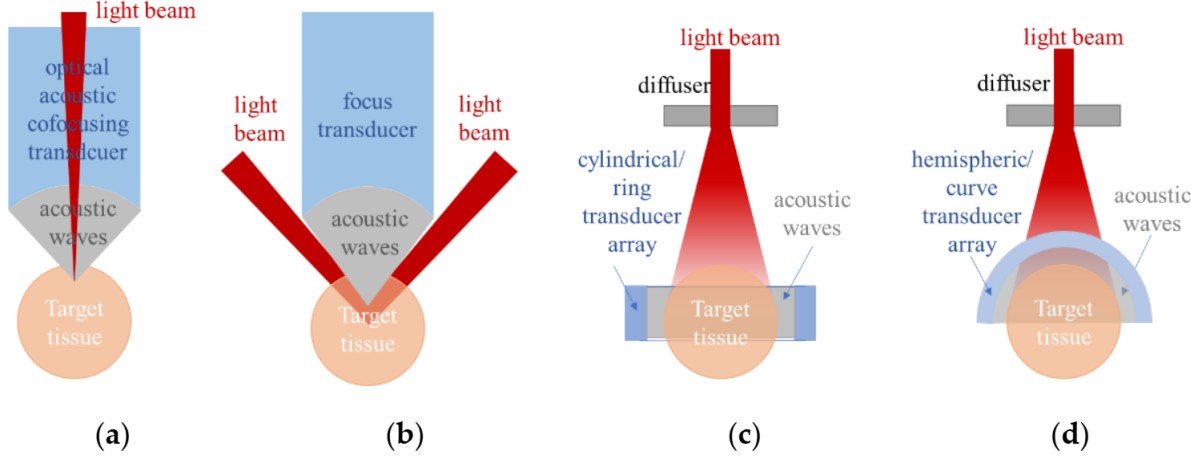

**Figure 1.** Schemes of PAI techniques for tissue imaging, classified according to spatial resolution and imaging depth: (**a**) OR-PAM, (**b**) AR-PAM and PAMac, (**c**,**d**) PACT.

### 1.2. Key Devices in PAI

The most important (and costly) component in a PAI device is the photoacoustic emission source, usually in the form of a pulsed laser. Important laser parameters in PAI applications are pulse energy, wavelength, pulse duration, pulse repetition rate and pulse stability. Different PAI methods require different laser energies. For example, OR-PAM, AR-PAM and PAT require laser devices with an output energy range of a few tens of micro-joules, a couple of hundred micro-joules and mill joules, respectively, in visible and near-infrared (NIR) wavelengths. OR-PAM requires a pulse duration shorter than 10 ns to satisfy the stress limitation condition of maximum PA generation in the biological cell scale. Using a single detector entails a raster scan of targeted samples, before image reconstruction. For real-time imaging, lasers with a high pulse repetition rate from kHz to MHz are required, depending on the imaging zone and spatial resolution. Although the effect of the pulse-to-pulse energy stability of PAI can be calibrated by a photodetector, an energy stability above 2–3% is recommended for a quantitative real-time measurement of chromophores in a tissue without fluence calibration [26,27].

Another key component in PAI is the ultrasonic (US) detector that is used to receive the PA signal. These detectors can be divided into two categories: piezoelectric transducers and optical sensors. Usually, piezoelectric transducers have a higher sensitivity and are more convenient for an imaging setup, while optical sensors are not disturbed by electromagnetic interference and have a wider response bandwidth, but lower sensitivity. Piezoelectric transducers are the most common type of US detector used in PAI, due to their high sensitivity, simple signal detection and sufficient bandwidth. Indeed, the bandwidth and central frequency of the piezoelectric transducer determine the axial resolution of PAM. OR-PAM typically employs a 50–75 MHz central frequency transducer [8,16], whereas AR-PAM has a lower frequency, such as 20–50 MHz [8,11,28]. For PAMac and PAT, the center frequency of the transducer or array should be as low as 1–10 MHz [8] for detecting lower frequency PA signals from deep tissues with less US attenuation.

In addition, PAI requires a scanning system, which can involve a single element, a line or a ring-array, with the system type determining the imaging speed and quality. PAM tends to use single element detectors with a focused or converged light beam in forward-view detection to achieve high lateral resolution. As the PA signal naturally provides one-dimensional (depth) information (signal arrival time multiplied by acoustic speed in tissue), 2D raster-scanning will produce a 3D PA image. Common scanning devices include mechanical scanners (stepping motor or DC motor), voice-coil scanners, optical scanners and MEMS-mirror scanners. A mechanical scanner has a low B-scan rate, but a large scanning range, while an optical scanner's properties are the opposite. A voice-coil scanner represents a negotiation between the scanning rate and range. A B-scan rate of

40 Hz/mm and a scanning range of up to ~10 mm [29] allows for the real-time imaging of small animals. However, a voice-coil scanner has a relatively low loading ability, and fast scanning requires the detection head to be light in weight, limiting its design and performance. Recently, a scanner based on a water-immersible microelectromechanical system (MEMS) has been developed for the simultaneous scanning of light beams and PA waves. MEMS can maintain optical and acoustic confocal alignment with fast scanning and, at a 400 Hz B-scan rate over a 3 mm range, as it has a large scanning range [30]. Moreover, a 2-axial water-immersible MEMS scanner can perform 2D scanning for 3D PAM imaging without a mechanical scanning head or sample, meaning the system is much more compact [31].

## 2. Achievements of PAI Techniques in the Past Five Years

A typical OR-PAM has an optical-acoustic confocal system (as shown in Figure 1a), in which the optical beam focus point overlaps the transducer focus range to achieve optimum lateral resolution and detection sensitivity. This confocal system has two glass prisms [29–31], maintaining coaxial aligning of the exciting light beam and accepted PA waves. As the two glass prisms must be immersed in coupling water for acoustic detection and scanning, an acoustic impedance mismatch can be observed at the glass/water boundary, resulting in a lower transmission speed when a PA wave moves across the boundary before detection. To avoid this, a novel OR-PAM setup has been developed, with the two prisms removed, as shown in Figure 2 [32]. In the reported setup, the exciting laser pulse was passed through a polyvinylidene difluoride (PVDF) transducer with a hollow core and was reflected to a mouse brain using a home-made MEMS scanner, which also reflected PA signals from the brain to the PVDF transducer. Although the PVDF transducer had a lower sensitivity than the more commonly used lead zirconate titanate (PZT) ceramic transducers, avoiding the mismatch at the prism-water boundary resulted in a ~50% increase in overall PAM detection sensitivity. It must be mentioned that PVDF transducers often offer a broader bandwidth than PZT transducers, improving the axial resolution of PAM. The new functional PAM has achieved high-sensitivity volumetric imaging using a 1 MHz one-dimensional imaging rate with a 2.7 μm lateral and a 30 μm axial resolution, revealing capillary-level vascular dynamics. Its detection sensitivity can probably be further improved by using a PZT ceramic transducer with a hollow core to replace the PVDF transducer. This is because a PZT has a much higher piezoelectric effect than a PVDF, although PVDF offers an improved acoustic coupling and transmission efficiency in water.

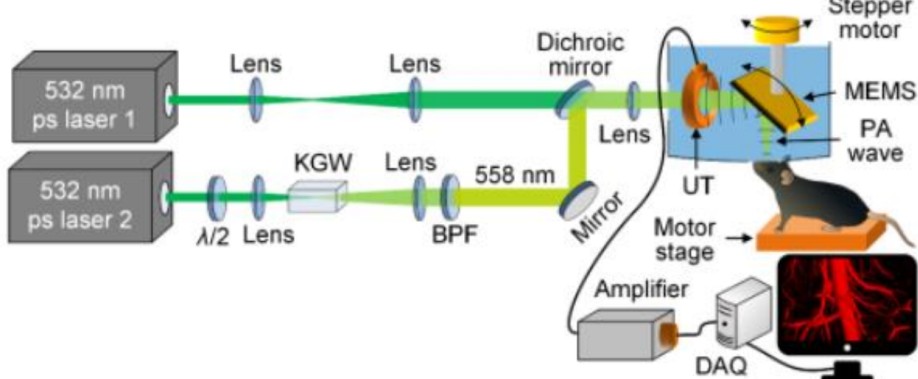

**Figure 2.** Schematic of a functional OR-PAM system without acoustic impedance mismatch: KGW, KGd (WO4) 2 crystal; λ/2, half-wave plate; BPF, band-pass filter; DAQ, data acquisition unit; and UT, US transducer (Reprinted from Ref. [32]).

In PAI, solid-state lasers are commonly used as illumination sources in tomographic or macroscopic imaging, due to their high pulse energy output (~10–100 mJ). With a pulse

duration in the nanosecond range, these properties are essential for highly efficient PA generation. However, their bulky size (sub-meter), heavy weight (a few tens of kg), high cost (>20 k€), and the relatively low pulse repetition frequency (~<100 Hz) mean solid-state lasers are unsuitable for clinical and portable applications. Over the last fifteen years, researchers have started to use high-power pulse laser diodes (PLDs) as an illumination source, largely because laser diodes are smaller in size (centimeters), lighter in weight (~grams), less expensive (~100 s €) and have a high pulse repetition frequency (~10 kHz). However, a key limitation of pulsed laser diodes in PAI is their low peak output power (~<20 μJ@100 ns), especially in the nanosecond pulse duration range (~μJ@10 ns). Fortunately, by combining more than ten laser diodes at the 905 nm wavelength, it is possible to obtain PAI with similar SNR at the depth of 1 cm as with a 10 mJ Q-switched Nd: YAG laser, provided that the stress confinement condition is fulfilled [33]. However, this requires sacrificing the lateral resolution to some extent [34]. As a result, PLDs have been used exclusively for PAI of superficial vascular structures. Recently, the Quantel-Laser company has introduced high-power laser diode illuminators with greatly improved output pulse energy (on the order of mJ @ 30 ns) at a pulse repetition frequency of several kHz, enabling PAT imaging of deep brain areas in mice or rats. Based on one of these illuminators, Upputuri and Pramanik developed a PLD-PAT system with a spatial resolution of 384 μm. Their pulse diode illuminator possesses a wavelength of 803 nm and a single-element piezoelectric transducer with a center frequency of 2.25 MHz (70% fractional bandwidth) to circularly scan and accept PA signals produced in samples [35,36]. The system is capable of dynamic in vivo imaging within 5 s with a high signal-to-noise ratio (SNR) of ~48, while preserving the skin and skull. This is useful for the non-invasive study of neurofunctional activities as well as the characterization of pharmacokinetic and biodistribution profiles in the brain. Since then, Upputuri and Pramanik have modified their PLD-PAT system using an unfocused ultrasonic transducer (UST) with a 5 MHz center frequency to detect venous sinus distension in vivo, which is caused by intracranial hypotension (IH) in small animals. The IH was induced through a 45 μL cerebrospinal fluid (CSF) extraction, using a needle connected to a draw syringe. Five minutes after CSF extraction, an increase of ~30% was observed in the sagittal sinus area with a width of 40% ± 5% (from 493 μm to 677 μm) [37].

Pulsed light emitting diodes (LEDs) have recently made an appearance in PAI as illuminating sources [38], particularly for low cost, portable/hand-held devices. Differing from solid-state lasers and laser diodes, LEDs are non-coherent and eye-safe devices, making them more applicable in the clinical setting. On the other hand, their lower optical energy and greater divergence relative to PLDs, renders pulsed LEDs more challenging for PAI. Last year, a theoretical simulation of LED-based PAI showed that brain tumour margin vessels can be imaged for the safe resection of glioma [39], but there has not yet been a report for in vivo brain imaging using LED-PAI. However, with the technical progress of LED devices, the authors believe that LED-based PAI can become a potential tool for preclinical and clinical brain imaging applications. Aside from brain imaging, other LED and PLD-based PAI applications can be found in the referenced literature [40–42].

In the past few years, fiber lasers have also been applied to PAI, OR-PAM in particular, as excitation sources for PA generation [43]. Compared to solid-state lasers, fiber lasers prove to be more compact and offer higher efficiency without an external water-cooling system. In comparison to laser diodes and LEDs, fiber lasers provide considerably better beam orientation and easier focusing. Moreover, fiber lasers have a high pulse repetition frequency (up to MHz), making them suitable for the construction of ultra-fast OR-PAMs [44,45]. Furthermore, based on their good beam quality and high pulse energy, fiber lasers, combined with a stimulation Raman frequency-shifting fiber, can produce wavelength-selectable optical sources [46], that are well-suited for building multi-wavelength OR-PAM systems [14,47]. As of relatively recently, Allen et al. have developed a high-pulse energy fiber laser, capable of providing an output energy of ~10 mJ with a variable pulse duration range of 10 ns–500 ns and a pulse repetition frequency of up to

1 kHz [48]. This compact fiber laser offers a practical excitation source for a portable PAT, which could be used for deep brain imaging.

The development of wearable PAI has been ongoing since 2015, when the first wearable PAT device was mounted on a rat's head while the animal was awake and moving around. Compared with rats under anaesthesia, it was observed that its cerebral blood volume (CBV) changed at a quicker rate over time, and was larger in amplitude for rats in a fully awake, moving state. This PAI setup achieved a lateral resolution of ~0.5 mm and an axial resolution of ~0.1 mm at a depth of up to 11 mm in phantom evaluation. It was capable of imaging blood vessels to a depth of 5 mm, maintaining an intact scalp and skull bone [49]. The researchers then modified the setup by using three layers of fully functional acoustic transducer arrays to build a three-dimensional wearable PAI system (3D-wPAT). Moreover, the in-plane spatial resolution was improved to 200 μm and acquired a 3D brain image within 167 ms. The experiment revealed that 3D-wPAT can detect visually evoked responses in the primary visual cortex and map cerebral oxygen saturation via multi-wavelength irradiation in behaving hyperoxic rats [50]. Further, Chen et al. reported a wearable OR-PAM mounted on a rat head, in which the imaging probe weighed only 8 g, allowing the rat to move around freely. The lateral and axial resolution of the acquired PA images were 2.25 μm and 105 μm, respectively, and only 10 s were required to image an area of 1.2 mm × 1.2 mm [51]. This wearable OR-PAM successfully monitored both morphological and functional variations in the vasculature of the cerebral cortex during the induction of ischemia and reperfusion. Last year, a fully portable and wearable PAM was reported [52], allowing a mouse to move around freely without the hindrance of any external devices connected via a cable or fiber (WiFi or wireless connection, as shown in Figure 3). Although the report does not contain in vivo data, displaying only phantom study results, wearable devices with a wireless connection are being used more often.

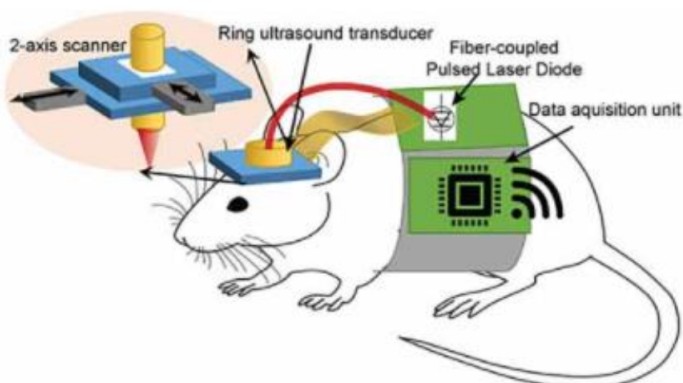

**Figure 3.** Illustration of a wearable PAM for mouse brain imaging (Reprinted with permission from Ref. [52]. 2020, IEEE).

Wearable PAI devices for the human head are more complicated to implement than those designed for small animals. Their practical realization typically suffers from the complex assembly, unviability of full-hat rotation around the human head, US coupling and the necessity of hundreds of US data acquisition channels required to cover the whole brain. Last year, a modular PAI hat was developed for neonatal brain imaging, consisting of multiple PAI disc modules, measuring 2 inches in diameter [53]. Conforming to the shape of the head, these modules were assembled onto a hat to cover the entire neonatal brain. Each of the PAI discs integrated source optical fibers and either densely packed US elements (to eliminate the need for rotation) or fewer US elements (usually in a trapezoidal shape) on a rotating disc (to reduce the number of data acquisition channels). Through an innovative modular design, the system minimized challenges posed by backend electronics.

Most high-resolution optical imaging of small animal brains has been carried out under anaesthesia to decrease motion artifacts. However, there is direct evidence that brain activity during wakefulness cannot be reliably inferred from the observations of

anesthetized animals [54]. Moreover, anaesthetic-induced changes in cerebral hemodynamics have been reported in humans and primates [55,56]. To solve this problem, Cao et al. developed a so-called head-restrained PAM system, in which the head of a fully awake mouse was restrained by a nut-bolt configuration [57]. In this configuration, a small nut was attached to the exposed mouse skull using dental cement. Fixed in a customized head plate with a bolt, the mouse head could be angularly adjusted using a rotation mount to align the region of interest perpendicular to the imaging head. It could be vertically adjusted using a right-angle clamp to allow the mouse's limbs to comfortably rest on the surface of a spherical treadmill. The treadmill consisted of two hollow polystyrene hemispheres, 8 inches in diameter, situated in a homemade cylindrical holder. The slightly compressed air from the bottom of the holder created a thin cushion to float the treadmill, allowing the mouse to move freely with a reduced reaction force. The head-restrained PAM enabled a simultaneous imaging of its cerebrovascular anatomy, total concentration and oxygen saturation of its hemoglobin and blood flow. A side-by-side comparison of awake and anesthetized brains revealed an isoflurane (a general anaesthetic) induced diameter-dependent arterial dilation, elevated blood flow and reduced oxygen extraction fraction in a dose-dependent method. As a result of the combined effects, the cerebral metabolic rate of oxygen was significantly reduced in the anesthetized brain under both normoxia and hypoxia, suggesting it to be a mechanism for anaesthetic neuroprotection.

The technical advances also include studying optical fluence distribution in the brain. Conventional PACT either assumes homogenous optical fluence within the brain or uses a simplified attenuation model for its estimation. Both approaches underestimate the complexity of fluence heterogeneity which can result in poor quantitative imaging accuracy. Tang and Yao employed a 3D Monte Carlo (MC) simulation to study optical fluence distribution in a complete mouse brain model [58]. They applied the MC simulation package on a digital mouse (Digimouse) brain atlas with complete anatomical information. Simulation results clearly illustrate that optical fluence in the mouse brain is heterogeneous at the global level and can decrease by a factor of five with increasing depth. Moreover, strong absorption and scattering in the brain vasculature both induce fluence disturbances at the local level. Correcting the optical fluence distribution can improve the quantitative accuracy of PACT. Another promising advancement in brain imaging involves a deep learning application for virtually increasing permissible exposures in order to enhance the signal-to-noise ratio in the deep structures of brain tissue. An efficient fully convolutional network (FCN), known as the U-Net, was trained using a perceptually sensitive loss function to understand how to enhance low-SNR structures [59]. An evaluation of the proposed method was conducted through sheep brain imaging experiments in vivo. As shown in Figure 4, the results indicate that the deep learning kernel increased the peak-to-background ratio by 5.53 dB and the penetration depth by 15.6%, compared to low-energy generated PA images. This method can facilitate the clinical translation of PA techniques to brain imaging, especially for transfontanelle brain imaging in neonates.

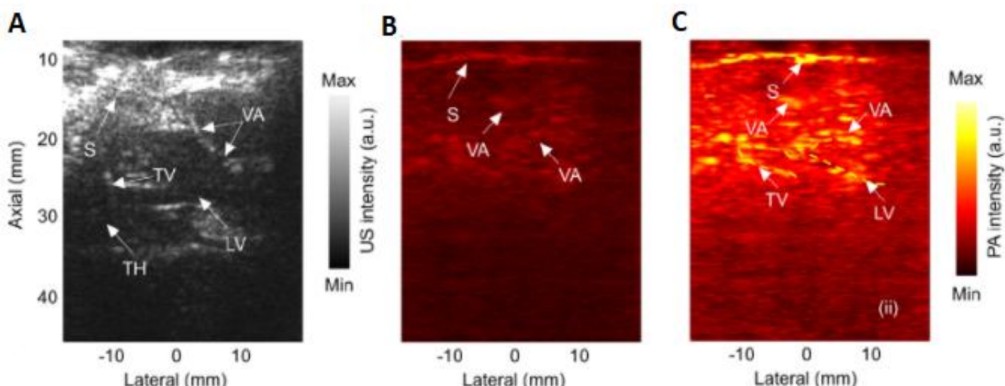

**Figure 4.** Performance evaluation of trained U-Net kernel based on in vivo sheep brain images. (**A**), US image along the sagittal plane. (**B**), PA image generated with a low energy laser; (**C**), deep learning enhanced PA image in (**B**). S, skin; LV, lateral ventricle; TV, third ventricle; TH, thalamus; VA, vasculature. Yellow contours indicate some of the structures that were not present in B (Reprinted with permission from Ref. [59]. 2020, Wiley-VCH).

## 3. Exogenous Reagents for PAI

Similar to other techniques, in deep tissue absent of endogenous absorbers or where the absorbing coefficients are low, the use of PAI can increase imaging contrast in deep tissue without endogenous absorbers or a sufficiently strong absorbing coefficient. This occurs through the introduction of exogenous absorption materials, such as small molecule dyes, metallic nanoparticles, single-walled carbon nanotubes, semiconducting polymer nanoparticles and quantum dots in the interested location. If the exogenous contrast agents can be bio-conjugated and attached with specific cells or tissues, they can also enable targeted molecular imaging, in addition to imaging contrast enhancement. Hence, these exogenous agents may greatly extend the preclinical and clinical application of PAI and other resembling techniques. Details of the exogenous contrast agents for PAI enhancement can be found in recent review papers [24,60], here we focus on the progress made in brain imaging using contrast agents over the past five years.

Deep imaging of brain and body tissues using PAI requires NIR wavelengths due to the minimal optical attenuation caused by low tissue absorption/scattering. However, low tissue absorption results in a weak PA generation. Moreover, the intrinsic biomarkers of many diseases cannot generate PA signals in the NIR range. This means it becomes necessary to develop exogenous PA contrast agents to improve the SNR of PAI in deep tissues and/or the accurate detection of diseases in living systems. Conjugated polymer nanoparticles (CPNs) are increasingly attracting interest as PAI contrast agents, due to their inherent advantages, which include a large absorption coefficient (compared with carbon nanotubes), excellent photostability (better than gold nanorods) and good biocompatibility. Recently, Liu et al. [61] rationally designed three donor–acceptor-structured conjugated polymers and prepared the corresponding CPNs. The chemical structures of these CPNs were nearly identical, except for the heteroatom of the acceptor moiety, which was systematically varied from oxygen to sulphur to selenium. The results indicate that the absorption coefficient, and in turn the PA signal intensity, decreased when the heteroatom changed from oxygen to sulphur to selenium. As these CPNs exhibit a weak fluorescence and a similar photothermal conversion efficiency (≈70%), their PA intensities were approximately proportional to their absorption coefficients, which was demonstrated by in vivo brain vasculature imaging. The experiment also evidenced that CNPs offer good photostability and do not have toxicity.

Research has suggested that molybdenum disulphide (MoS2) nanosheets have a good physiological stability and biocompatibility, higher optical absorption than gold nano-rods and graphene derivatives in the NIR wavelength, and excellent enhanced permeability and retention (EPR) effects for tumour targeting. As a result, Chen et al. investigated MoS2

nanosheets using differently layered nanostructures and observed that single-layered MoS2 presents a better PAI sensitivity than its multiple layered counterparts, due to its higher NIR absorption and superior elastic properties [62]. An intravenous administration of S-MoS2 to both U87 subcutaneous and orthotopic tumour-bearing mice revealed that a tumour located 1.5 mm below the skull can be clearly rendered by PAM in vivo. Later, the team developed a covalent conjugating strategy for the synthesis of a MoS2–indocyanine green (ICG) hybrid, which significantly enhanced the PAI sensitivity compared to MoS2 nanosheets [63]. Moreover, the absorbing peak was redshifted from 675 nm to 800 nm, resulting in a greater penetration depth and lower background noise. In vivo PAM of orthotopic brain glioma demonstrated that MoS2–ICG has is capable of clearly identifying a tumour mass positioned 3.5 mm below the scalp, which is almost twice the depth achieved using S-MoS2 nanosheets. This is one of the greatest depths reported among all glioma PAM studies so far by using a nanoprobe in the NIR I spectral region (700–100 nm).

Meanwhile, Guo et al. [64] designed and synthesized conjugated polymer nanoparticles, which possesses strong optical absorption in the NIR II (100–1700 nm) 1064 nm wavelength. They used benzobisthiadiazole to copolymerize with Benzodithiophene to form poly(benzodithiophene-alt-benzobisthiadiazole) (P1) via Stille polymerization. The absorption peak at the 1064 nm wavelength of P1 nanoparticles (NPs) is suitable for PA generation when using common and low-priced Nd: YAG pulsed lasers. Moreover, the wavelength is in the NIR II range, which possesses less optical absorption and attenuation, but more penetration depth in skull and brain tissues compared with the light in NIR I, meaning it is better suited for deep imaging in the brain. In vivo PA imaging revealed that an orthotopic tumour range as deep as 3.4 mm can be clearly imaged in mice brains after an injection of P1 nanoparticles, and that 24 h after injection, the PA signal of the tumour had, in this instance, increased 94-fold. It reveals that P1 nanoparticles have a good biocompatibility, excellent photostability, high imaging contrast and signal/background ratio, passes the blood brain barrier (BBB) successfully and that it accumulates in tumours. Afterwards, the research group developed cyclo(Arg-Gly-Asp-D-Phe-Lys (mpa)) decorated P1 nanoparticles (P1RGD NPs) [65], with an identical absorption spectra to P1 nanoparticles. P1RGD NPs have a layer of c-RGD peptides covalently decorated on the P1NPs' surface to enhance NP uptake inside a tumour. In combination with the 1064-nm laser illumination, the P1RGD NPs enabled dual PA imaging and photothermal therapy in deep brain areas through scalp and skull, using the necessary optical power below the laser safety limit. Meanwhile, the group also developed a NIR-II fluorescent molecule with aggregation-induced-emission (AIE) characteristics for the imaging of orthotopic brain tumours. An encapsulation of the molecule in a polymer matrix yielded AIE dots, showing a very high quantum yield of 6.2% with a large absorptivity of 10.2 $cm^{-1}/gL^{-1}$ at 740 nm and an emission maximum of almost 1000 nm [66]. These AIE dots can also be decorated with c-RGD to enable the specific and selective tumour uptake. Hence, this method presents great potential for precise dual NIR-I PAI and NIR-II fluorescence diagnostics of deep brain tumours through the scalp and skull. Recently, the group fabricated other conjugated polymer NPs, which showed an even higher absorptivity of 35.2 $cm^{-1}/gL^{-1}$ at 1000 nm and an emission peak at 1156 nm [67]. These NPs were used to build a dual NIR-II PAI and fluorescence system that allowed for the mapping of small tumours measuring less than 2 mm at a depth of 2.4 mm beneath the scalp and skull. This method had a large signal-to-background ratio after BBB opening, and simultaneously resolved hemodynamics and cerebral vasculatures with a spatial resolution of 23 μm at a brain depth of 600 μm.

For the imaging of deeper brain gliomas, Liu Y. et al. [68] designed and synthesized a mesoionic dye A1094, which was encapsulated in an Arg-Gly-Asp-modified hepatitis B virus core protein (A1094@RGD-HBc), with an aggregation-induced absorption enhancement (AIAE) mechanism in the NIR II window. Differing from conventional chromophores, such as gold NPs for which aggregation usually leads to a reduction of imaging contrast due to the quenching effect, the AIAE absorbers exhibited excellent photostability. The experiment demonstrated that A1094@RGD-HB displayed a ninefold PA signal amplification

in vivo and allowed the precise imaging of brain gliomas at a depth of more than 6 mm in the NIR II window.

Suitably assembled tumour-targeting nanoparticles can be used in multifunctional phototheranostics to enhance the synergistic effect of photo-thermal therapy (PTT) and PA therapy under the guidance of PAI [69,70]. In the reported study, nanoparticles selectively penetrated the BBB within the tumour range and accumulated in the range. These nanoparticles converted the pulsed laser energy into a shockwave via PA cavitation to achieve localized mechanical damage and thus yielded a precision antitumour effect. In addition to their therapeutic function, nanoparticle-mediated PA processes can also generate images that provide valuable information regarding tumour depth, size and vascular morphology to inform the treatment design and its monitoring [70]. The results revealed that glioblastoma tumours were selectively destroyed without any observable side effects on the normal tissue.

Naturally, it is easier for small-molecule PA probes to cross the BBB for diagnostics and imaging. Wang et al. [71] developed a series of activatable PA probes (RPS1-RPS4), which were specifically chelated with $Cu^{2+}$ to form radicals with turn-on PA signals in the NIR region, where RPS1 displayed a fast response (within seconds) to $Cu^{2+}$ with high selectivity and a low PA detection limit of 90.9 nM. Owing to its low molecular weight and effective BBB crossing ability, RPS1 therefore lends itself to the PA imaging of Alzheimer's disease in mice brains, because copper enrichment in the brain is highly related to AD pathogenesis.

PAI has also been used to measure neuronal voltage response signals, which are closely related to epilepsy, in live mouse brains. Rao et al. used a spectroscopic PAT at two isosbestic points of hemoglobin at 500 nm and 570 nm to separate voltage response signals and hemodynamic signals on live brain surfaces through a non-radiative voltage dye (dipicrylamine). As was observed from the in vivo measurements before and during 4-Aminopyridine induced epileptic seizures, voltage response signals are much higher than hemodynamic signals [72]. PAI also demonstrated human embryonic kidney 293 (HEK-293) cell membrane voltage responses in rat brain tissue at 4.5 mm, which is beyond the optical diffusion limit. This imaging depth could be further improved on through the application of NIR non-radiative voltage dyes. Kang et al. used a NIR cyanine voltage sensitive dye (VSD, IR780 perchlorate) combined with functional PAI to monitor electrophysiological neural activities in real-time [73]. The VDS was delivered through the BBB to a rat brain. The results demonstrated that functional PAI of the fluorescence quenching VSD mechanism proved to be a promising approach to recording the brain activities in a chemoconvulsant rat model, at a sub-mm spatial resolution, with an intact scalp. The group also reported on that the transcranial functional PAI of N-methyl-D-aspartate (NMDA) evoked neural activity in the rat hippocampus [74]. Voltage-sensitive dye (VSD) contrast was used to identify neural activity changes in the hippocampus, which correlated with the NMDA-evoked excitatory neurotransmission. The imaging results revealed a positive correlation between VSD responses and extracellular glutamate concentration changes. Hence, the technique was capable of distinguishing focal glutamate loads in the rat hippocampus, based on the VSD redistribution mechanism, which is sensitive to the electrophysiologic membrane potential.

The imaging and rehabilitation of brain injuries are important aspects in regenerative medicine, where dynamic visualization remains a challenge due to imaging technology limitations and/or cell labelling difficulties. PAI has been used in combination with a highly efficient NIR dye (citrate-coated Prussian blue) to noninvasively monitor traumatic brain injury (TBI) and its rehabilitation [75]. Proving to be a good PA contrast agent, the dye labelled bone mesenchymal stem cells (BMSCs) was intravenously injected in mouse tails for an improved observation. PAI was successfully used to monitor the labelled BMSCs, which identified the TBI region and repaired the ruptured vasculature faster than without using BMSC treatment.

Focus ultrasound (FUS) is a powerful technique used to open the BBB temporarily and reversibly, which is of interest for treating neurodegenerative diseases and tumours

through the enhancement of drug delivery. However, the detection of BBB opening via non-invasive methods remains limited. When combined with suitable fluorophore-labelled NPs or with indocyanine green (ICG), PAI can be used to detect BBB openings [76]. The spectral unmixing of PA images of in vivo (2 h post FUS), perfused and ex vivo brains revealed a consistent distribution pattern of ICG and NPs at FUS locations. The spectrally unmixed ex vivo PA images demonstrated that both the opening width and spread laterally correlated well with BBB opening locations on MR images. The paper also demonstrated that a 3D colour Doppler can be used to monitor BBB opening over time by detecting reversible anatomical changes after FUS treatment.

Other exogenous reagents with the capacity to simultaneously enhance imaging/sensing contrasts in multimodal techniques can be found in the section below.

## 4. PAI in Combination with Other Imaging Techniques

Both multimodal imaging and detection are increasingly used in practical applications to ensure an enhanced accuracy and/or to attain more comprehensive answers to targeted questions. Liu et al. designed and synthesized a multifunctional probe using a NIR ultra-high absorbing croconium dye for amyloid (CDA), which can be used to strongly bind the hydrophobic channels of cerebrovascular amyloid beta (Aβ) fibers. Labelled with radioactive 18F, the multifunctional probe allowed for the ultrasensitive PAT/positron emission tomography (PET)/fluorescence imaging (FI) of Aβ plaques in the brain cortex [77]. PAT has been used in multimodal detection to image pathological sites on cortical vessels with a high spatial resolution and optical specificity, whereas PET has revealed a whole-body anatomy with quantitative biodistribution information. PAT has also been employed in combination with fluorescence molecular imaging (FMI) to non-invasively detect tumours through an intact skull, by combining targeted cancer stem cells nanoparticles. These nanoparticles accumulated specifically in the tumour area, allowing glioma to be precisely revealed at the cellular level [78]. This combination of optical PAT and FMI navigation fulfilled the promise of precise visual imaging in glioma detection and the resection in deep tissue.

Mesenchymal stem cells (MSCs) have been used in a variety of successful applications against glioblastoma multiform (GBM) identifying brain tumours. A recent development involves gold-coated superparamagnetic iron oxides (SPIO) loaded with bone marrow-derived MSC nanoparticles (SPIO@Au-loaded MSCs) which were used in PA and magnetic resonance (MR) dual-modal imaging [79]. The synthesized SPIO@Au-loaded MSCs function as contrast enhancing agents for PA and MR, because gold nanoshells produce surface plasmon resonance in the NIR region, while SPIO nanocores produce strong MR signals through the skull. The experiments revealed that carotid artery-injected MSCs labelled with SPIO@Au at 4 μg/mL did not induce cell death or any adverse effects on differentiation or migration, and could be tracked in vivo via MR and PA imaging. This provides a powerful tool for the design and real-time monitoring of stem cell-mediated brain tumour therapies. It is also worth noting that gold nanoshells on nanoparticle surfaces permitted the use of laser ablation to kill cancer cells, as in photothermal ablation therapy.

Based on the research above, it appears integrating the nanocomposite design with the multimodal imaging capabilities of PAI, FI and MR imaging (MRI) offers multiple advantages for the sensitive imaging of early-stage glioblastoma tumours, while compensating for the limitations of each imaging modality. Recently, Duna et al. developed multifunctional nanoagents, based on NIR light-absorbing and NIR-emissive small molecules (TC1), and integrated them with ultra-small iron oxide nanoparticles (UIONPs) [80]. To resolve the fluorescence quenching problem induced by the immediate proximity of UIONPs and the TC1, they fabricated HALF (which is the nanocomposite volume to be half-occupied by UIONPs) to create spatial distance between the two components. Decorated with a peptide ligand cRGD for improved brain tumour targeting, HALF-cRGD was demonstrated, experimentally, to be a good imaging contrast agent in multimodal imaging (FI, PAI and MRI) and an effective photothermal therapeutic agent for brain tumours.

Raikwar et al. used multimodal techniques, combining real-time non-invasive bio-luminescence imaging (BLI), high-resolution in vivo PAI and neuropathological analyses to provide an accurate validation of traumatic brain injury (TBI) pathology, by subjecting NFκB-RE-Luc transgenic male and female mice to TBI (Model #10,499, Ta-conic Biosciences, Rensselaer, NY, USA) [81]. Experimental results showed that real-time noninvasive PAI, in combination with other neuroimaging modalities, successfully produced a rapid TBI diagnosis, the rapid monitoring of TBI dynamics and long-term longitudinal preclinical monitoring of precision-targeted TBI therapies.

Dual-mode US and PAI imaging is increasingly used in preclinical applications because they provide both anatomical and functional information. Combined with PAI with contrast enhancement agents and delivered to the brain by FUS BBB opening, the capabilities of dual-mode imaging can be extended to the real-time visualization of molecular properties and drug delivery in the brain at a depth of several centimeters without craniotomy. Kartman et al. developed a multiple-mode US-guided PAI technique using the commercial Vevo LAZR PA apparatus (VisualSonics, Inc., Toronto, ON, Canada) [82]. Their system delivered silica-coated gold nanorods with strong NIR absorption characteristics to the brain, via a microbubble-assisted FUS BBB opening. These nanorods functioned as an imaging contrast enhancement agent for longitudinal diagnostic imaging and for the therapeutic monitoring of neurological diseases. Therefore, the method has the potential to act as a highly useful tool for understanding the mechanisms of neurological disorders and for evaluating appropriate treatment choices.

Devising a method for non-invasive PA brain imaging, which is different from PA body tissue imaging, has proved challenging, largely due to the skull bone, which has a considerable effect on acoustic attenuation, aberration and reverberation. There exists a considerable disparity between acoustic impedance in the skull bone and in soft brain tissue. As a result, before propagating outside the skull, PA waves experience great attenuation, acoustic mode conversion between longitudinal and shear waves and wave reverberations at the two skull boundaries. These disadvantages complicate accurate image reconstruction. Yao's group systematically investigated the impact of murine skull on PA wave propagation and the corresponding PAI image reconstruction [83]. Their results confirmed that transcranial PACT displayed an inferior image quality compared with transcranial PAM (seen Figure 5), even in instances where the two modalities use a similar US frequency and effective detection aperture. Moreover, they established that phase aberration, which is caused by different longitudinal wave speeds in the skull and the surrounding soft tissues is a major reason for the transcranial PACT's image artifacts. One potential solution is to develop an adaptive image reconstruction method that can incorporate inhomogeneous wave speeds to numerically reduce the phase aberration. This method typically benefits from acquiring prior information regarding the skull's geometry and acoustic properties, which can be determined using the US technique. Phase aberration can also be reduced by chemically softening the skull to reduce longitudinal wave speeds. Another major contributor to image artifacts is mode conversion at skull boundaries. However, Yao's group found that direct transmission of shear wave-converted longitudinal waves was less affected by distortions introduced by the skull and resulted in an improved reconstructed image quality with less artifacts. Wave reverberations, on the other hand, primarily caused image artifacts in deeper brain regions, where the useful signals (or directly transmitted signals) are often weak too. Reverberation artifacts may be reduced by applying low-pass filtering before the image reconstruction, because reverberation waves follow a longer acoustic path length within the skull and possess more high-frequency components than directly transmitted waves. Other methods include the application of a first-of-a-kind dictionary learning algorithm [43], or the removal of late-arrived signals in half-time image reconstruction [84].

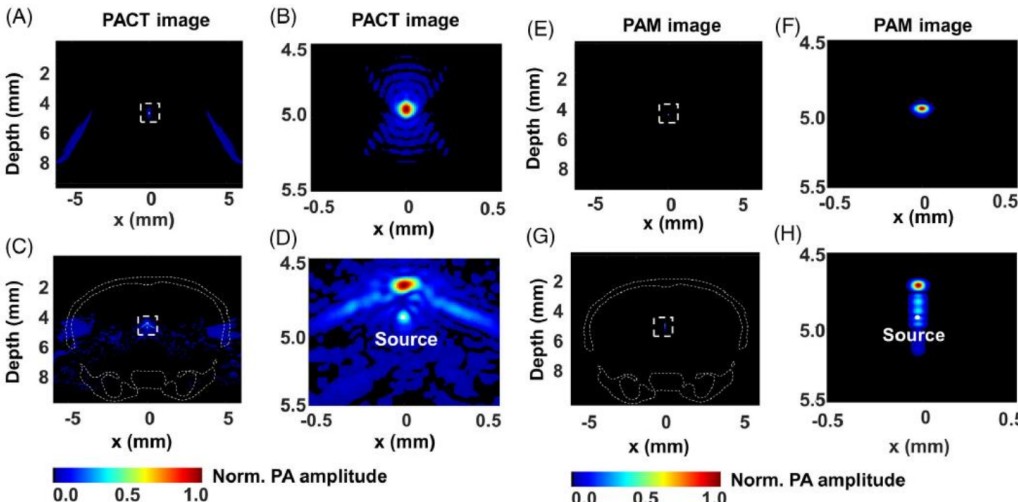

**Figure 5.** Simulated transcranial PAM and PACT images of a point target in water and skull cavity. (**A**) Full PACT image of the point target in water. (**B**) A close-up image of the region indicated by the dashed box in (**A**). (**C**) Full PACT image of the point target in water within the skull. (**D**) A close-up image of the region indicated by the dashed box in (**C**). (**E**) PAM image of the point source in water. (**F**) A close-up image of the region indicated by the dashed box in (**E**). (**G**) Full PAM image of the point target in water enclosed by the skull. (**H**) A close-up image of the region indicated by the dashed box in (**G**). The white dots in (**D,H**) represent the correct location of the point target (Reprinted with permission from Ref. [83]. 2019, Wiley-VCH).

## 5. Conclusions and Outlook

The application of PAI to biological and biomedical research has made great strides in the past twenty years. In recent years, PAI has been brought to bear on small animal brain imaging. As for PA devices, technological development is in rapid progress and the costs involved are gradually decreasing. Fiber lasers are emerging with higher pulse repetition rates and a more convenient wavelength tuning. Several groups have developed LD-PAI and LED-PAI setups that are relatively low-priced and less bulky than before. Wearable, and even wireless PAI setups, have been introduced for brain imaging, capable of observing the behaviour and activities of unanaesthetised mice, allowing for the study of the brain functions and metabolism in awake and moving subjects. This presents a new avenue for PAI-based monitoring of behaviour-related neuro and hemodynamic activities.

Exogenous contrast enhancement reagents in PAI will become increasingly important for monitoring neural activity, glucose uptake, aberrant protein aggregation, malignancy, deep brain tumours, traumatic brain injury, BBB openings, epilepsy and Alzheimer's disease. Over recent years a frequent application of multi-modal imaging techniques in preclinical research has been witnessed. Examples include PAI/US for lymph nodes [63,82], PAI/OCT (optical coherence tomography), PAI/EEG (electroencephalography) for epiepsy [85–87], PAI/PET for brain plaques [77], PAI/MRI/FL (fluorescence) [78,80] and PAI/MRI/computer tomography (CT) [79,88] for tumours and glioblastomas.

The skull effects arise from the acoustic speed and impedance mismatch between the skull and soft tissues, as well as from the layered structure of the skull (outer ivory-marrow-inner ivory). Speed mismatch causes phase aberration, which is the main source of artifacts in transcranial PAI. Current research shows that even a small animal skull with a thickness as thin as 0.3 mm will strongly distort PAI of the brain [83]. Three major factors that degrade the image quality of PACT are phase aberration, mode conversion and reverberation. PAM and PAMac, on the other hand, suffer mainly from signal reverberation, producing an image quality that is relatively improved and less affected by the skull. Reverberation artifacts can be easily decreased by applying low-pass filtering before the image reconstruction. However, the way in which the skull of a small animal affects the image quality may be greatly reduced by applying skull optical clearing solutions, which

enhances the transmission of exciting light and PA signals through the skull [89]. As the skull of a human is much thicker than that of small animal, the effect the skull has on transcranial PAI would be much stronger, lessening the quality of the reconstructed image. More advanced algorithms need to be developed to reduce the identified effects. Before this, imaging neonates for which fontanelles have yet to fuse together provides the next step forward for PAI.

**Author Contributions:** Z.Z. wrote most of the text, whereas T.M. revised, commented on and discussed its content with Z.Z. during the writing process. All authors have read and agreed to the published version of the manuscript.

**Funding:** This research was funded in part by the Academy of Finland, grant number 318347.

**Institutional Review Board Statement:** Not applicable.

**Informed Consent Statement:** Not applicable.

**Data Availability Statement:** Not applicable.

**Conflicts of Interest:** The authors declare no conflict of interest.

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
