# Peer review of "Recent Technical Progression in Photoacoustic Imaging—Towards Using Contrast Agents and Multimodal Techniques"

_applsci, doi:10.3390/app11219804_

Round 1

Reviewer 1 Report

This is an interesting review to illustrate the recent progress in the field. Different photoacoustic imaging (PAI) techniques have been discussed and the key devices have been reviewed. The technical advancement in the past five years as well as the exogenous reagents used in PAI and integration with other imaging techniques has been extensively discussed in the manuscript. The outlook in the future reflects the sharp vision and may help the researchers in the field. As such, it is recommended to be published in Applied Sciences.

Author Response

This is an interesting review to illustrate the recent progress in the field. Different photoacoustic imaging (PAI) techniques have been discussed and the key devices have been reviewed. The technical advancement in the past five years as well as the exogenous reagents used in PAI and integration with other imaging techniques has been extensively discussed in the manuscript. The outlook in the future reflects the sharp vision and may help the researchers in the field. As such, it is recommended to be published in Applied Sciences.

  • We thank the reviewer to read our manuscript and give positive comments.

Reviewer 2 Report

The subject of this review article is chosen extremely broad and cannot be covered
completely. I suggest a complete rewrite and resubmit. Such an article should not pick
seemingly disconnected research work from a broad subject but focus on one (sub-)subject.
This subject need to be clearly defined from the beginning. In the current form I see little
added value to existing review articles
Rather than being re-contextualized, the text passages are often obviously copied from
original articles – which I verified using a plagiarism check software. This is especially
apparent in figure 5 which is identical in content and caption to fig 8 from [74] with no
mention of reproduction or permission of reproduction. This is also relevant for figures 2-4.
Minor points include:
- introduce all abbreviations on first use, e.g. PAI US, this is especially important as this is
intended for a non-expert audience
- frequent typos: l.34 “]]”, l.41 “.” and odd hyphenation: “bio-logical”, “ultra-sound”, “imaging”
...
- why “brain” in figure 1?
- l. 86 ”high pulse repetition rate from kHz to MHz” PAI for human applications rep rates
are usually tens of Hz
- l. 90 unreferenced claim … and unclear what “quantitative” refers to
- l. 94 odd phrasing “immunize”

Author Response

The subject of this review article is chosen extremely broad and cannot be covered
completely. I suggest a complete rewrite and resubmit. Such an article should not pick
seemingly disconnected research work from a broad subject but focus on one (sub-)subject.
This subject need to be clearly defined from the beginning. In the current form I see little
added value to existing review articles
Rather than being re-contextualized, the text passages are often obviously copied from
original articles – which I verified using a plagiarism check software. This is especially
apparent in figure 5 which is identical in content and caption to fig 8 from [74] with no
mention of reproduction or permission of reproduction. This is also relevant for figures 2-4.

ANSWER:

  • Thank you for the valuable comments for our manuscript.
  • Yes, the title of the manuscript is broad, however, we narrowed it by reviewing only the last five years of its progress. This was chosen in order to highlight the overall (impressive) technical progression of PAI, since it is very rapidly developing. We think that would give nice overview for readers to better understand the new process.
  • In addition, this review tries to introduce the advanced key components in photoacoustic imaging devices and to concisely generalize the technical development direction in preclinical imaging, again, based on summarizing the data to be published only in past five years. This is helpful for the persons who have not yet high expertise/experience in utilising PA related techniques, in order to awake interest to start utilising such imaging devices. We think this is the added value to existing review articles.
  • Because the manuscript is written as a review paper, most data are come from the database of Web of Science in recent five years. Naturally, we used our own words to write the text, but if there are data or figures which come from refereed papers, we usually added the reference numbers and keep some original words or sentences (if the words or sentences are clear enough to understand) for avoiding unnecessary misunderstanding and also for respecting the authors. In this point, we think it is not necessary to create different words or sentences just only for distinguishing the words from the cited original papers. So, because figures 2-5 are cited from related references, we almost kept the related figure captions as we think they are clear enough. Importantly, we have permission (from authors or journal) for figures 2-5 and added the permission.

Minor points include:
- introduce all abbreviations on first use, e.g. PAI US, this is especially important as this is
intended for a non-expert audience

- We have now re-checked all text, giving full names of the abbreviations when first used them.  - frequent typos: l.34 “]]”, l.41 “.” and odd hyphenation: “bio-logical”, “ultra-sound”, “imaging”

- change it to [ ], delete “.” and all odd hyphenation in the text.
...
- why “brain” in figure 1?

-we use now “target tissue” for more general description.                                                                                           
- l. 86 ”high pulse repetition rate from kHz to MHz” PAI for human applications rep rates
are usually tens of Hz

-Yes, solid lasers usually have rep rate of ten of Hz. But some OPO lasers, laser diodes, and fiber lasers can be KHz to MHz of pulse rep rate. Here we discuss real-time imaging, hence the pulse rep rate of the lasers should be fast enough, from kHz up to MHz, as in references [29,34, 40,42].
- l. 90 unreferenced claim … and unclear what “quantitative” refers to

- “quantitative PA measurement of chromophores in a tissue” means that the concentrations of chromophores such as hemoglobin or blood oxygen saturation will be determined exactly. Now, we added two references.

 Hence, the absorption coefficient of the chromophore should be deduced from the photoacoustic signal amplitude or image which is closely related to laser fluence. So, the stability of laser fluence/energy should be high enough for accuracy measurement without calibration. This is important in real-time measurement. 2~3% is our recommended value, not referring from other authors. So, we change it in the text as “an energy stability better than 2~3% is recommended for a quantitative real-time measurement of chromophores in a tissue without fluence calibration”.
- l. 94 odd phrasing “immunize”

-we changed “immunize” to “is not disturbed by”   

Reviewer 3 Report

This manuscript reviewing technical progression in photoacoustic imaging is well organized regarding the subsections. The authors also indicated  studies in the detail way showing how the technical progressions have impact on technology based on photoacoustic effect. However, before the recommendation for publication, I have some concerns about this work, the authors are expected to address the following concerns:

*Some terms and concepts in some part are not clear to me. To illustrate; in the abstract part, the below sentence might mislead the reader about the resolution on account of the working principle of photoacoustic.

“””Hence, photoacoustic imaging has both advantages, i.e.,  high contrast from optical absorption and high spatial resolution from ultra-sound “””.  Actually, the high spatial resolution does not result from ultrasound, it depends basically on optical spot. Maybe, authors wanted to indicate “high spatial resolution, especially in the axial direction” I would suggest to use the terms in optoacoutic field, not the general terms for the manuscript.

As below paragraph from the manuscript, the authors indicated more specific sentences that are need to be support these sentences by the papers, and maybe in the manuscript, the authors already have them, but it should be cited in a particular way after claims, at least end of the paragraph, especially in the reviewer papers.

“Another key component in PAI is the US detector used to receive the PA signal. These 91 detectors can be divided into two categories: piezoelectric transducers and optical sensors. 92 Usually, piezoelectric transducers have a higher sensitivity and are more convenient for 93 an imaging setup, while optical sensors immunize electromagnetic interference and have 94 a wider response bandwidth, but lower sensitivity. Piezoelectric transducers are the most 95 common type of US detector in PAI, due to higher sensitivity, simpler signal detection 96 and sufficient bandwidth. Indeed, bandwidth and the central frequency of the piezoelec- 97 tric transducer determine the axial resolution of PAM. OR-PAM typically employs a 50 - 98 70 MHz central frequency transducer, whereas AR-PAM has a lower frequency, such as 99.”

30 - 50 MHz. For PAMac and PAT, the center frequency of the transducer or array should 100 be as low as 1-10 MHz for detecting lower frequency PA signals from deep tissues with 101 less US attenuation.

*In the part of key advances in PAI, I don’t see any references related fiber laser. In the other parts, even if the manuscript include some references, (such as Allen et al. etc) in the other parts, the references should be also in this part. The properties of excitation source in the field of photoacoustic have a great role on the different applications, in recently, they are fiber excitation source that can be mentioned (Yavas, S, Scientific reports6(1), 1-10). Additionally, as you see in the below sentence, it should be referenced, especially if someone indicate specific values, and studies. Please check whole paper for this problem.

“Although the effect of the pulse to-pulse energy stability of PAI can be calibrated by a photodetector, an energy stability better than 2~3% is necessary for a quantitative PA measurement of chromophores in a tissue. “

*Although the figure 1 illustrated schemes of PAI techniques with respect to spatial resolution and imaging depth in a feasible way, is there any specific reason why the authors selected the key point of the paper as the study of brain activities, particularly in optoacoustic field? I would suggest the authors to check the important reviews (for example, Karlas, A., Fasoula, N. A., Paul-Yuan, K., Reber, J., Kallmayer, M., Bozhko, D., ... & Ntziachristos, V. (2019). Cardiovascular optoacoustics: From mice to men–A review. Photoacoustics, 14, 19-30) to extend the applications and emphasize the reason why we need to improve or focus on the imaging technology compared to the other technologies for the study of brain injuries, or disease such as DOT or other applications. DOT should be mention in the manuscript, especially brain application and also mentioned some prominent studies based on combined DOT and PAT.

This manuscript reviewing technical progression in photoacoustic imaging is well organized regarding the subsections. The authors also indicated studies in a detailed way showing how the technical progressions have an impact on technology based on the photoacoustic effect. However, before the recommendation for publication, I have some concerns about this work, the authors are expected to address the following concerns:

*Some terms and concepts in some parts are not clear to me. To illustrate; in the abstract part, the below sentence might mislead the reader about the resolution on account of the working principle of photoacoustic.

“Hence, photoacoustic imaging has both advantages, i.e.,  high contrast from optical absorption and high spatial resolution from ultra-sound “.  Actually, the high spatial resolution does not result from ultrasound, it depends basically on the optical spot. Maybe, authors wanted to indicate “high spatial resolution, especially in the axial direction” I would suggest to use the terms in optoacoutic field, not the general terms for the manuscript.

As below paragraph from the manuscript, the authors indicated more specific sentences that are need to be support these sentences by the papers, and maybe in the manuscript, the authors already have them, but it should be cited in a particular way after claims, at least end of the paragraph, especially in the reviewer papers.

“Another key component in PAI is the US detector used to receive the PA signal. These 91 detectors can be divided into two categories: piezoelectric transducers and optical sensors. 92 Usually, piezoelectric transducers have a higher sensitivity and are more convenient for 93 an imaging setup, while optical sensors immunize electromagnetic interference and have 94 a wider response bandwidth, but lower sensitivity. Piezoelectric transducers are the most 95 common type of US detector in PAI, due to higher sensitivity, simpler signal detection 96 and sufficient bandwidth. Indeed, bandwidth and the central frequency of the piezoelec- 97 tric transducer determine the axial resolution of PAM. OR-PAM typically employs a 50 - 98 70 MHz central frequency transducer, whereas AR-PAM has a lower frequency, such as 99.”

30 - 50 MHz. For PAMac and PAT, the center frequency of the transducer or array should 100 be as low as 1-10 MHz for detecting lower frequency PA signals from deep tissues with 101 less US attenuation.

*In the part of key advances in PAI, I don’t see any references related to fiber laser. In the other parts, even if the manuscript includes some references, (such as Allen et al. etc) in the other parts, the references should be also in this part. The properties of excitation sources in the field of photoacoustic have a great role on the different applications, in recently, they are fiber excitation sources that can be mentioned (Yavas, S, Scientific reports6(1), 1-10). Additionally, as you see in the below sentence, it should be referenced, especially if someone indicates specific values and studies. Please check the whole paper for this problem.

“Although the effect of the pulse-to-pulse energy stability of PAI can be calibrated by a photodetector, energy stability better than 2~3% is necessary for a quantitative PA measurement of chromophores in a tissue. “

*Although figure 1 illustrated schemes of PAI techniques with respect to spatial resolution and imaging depth in a feasible way, is there any specific reason why the authors selected the key point of the paper as the study of brain activities, particularly in optoacoustic field? I would suggest the authors to check the important reviews (for example, Karlas, A., Fasoula, N. A., Paul-Yuan, K., Reber, J., Kallmayer, M., Bozhko, D., ... & Ntziachristos, V. (2019). Cardiovascular optoacoustic: From mice to men–A review. Photoacoustics, 14, 19-30) to extend the applications and emphasize the reason why we need to improve or focus on the imaging technology compared to the other technologies for the study of brain injuries, or disease such as DOT or other applications. DOT should be mention in the manuscript, especially brain application, and also mentioned some prominent studies based on combined DOT and PAT.

Author Response

This manuscript reviewing technical progression in photoacoustic imaging is well organized regarding the subsections. The authors also indicated  studies in the detail way showing how the technical progressions have impact on technology based on photoacoustic effect. However, before the recommendation for publication, I have some concerns about this work, the authors are expected to address the following concerns:

We thank the reviewer to read our manuscript carefully and give many comments for improving our paper.

*Some terms and concepts in some part are not clear to me. To illustrate; in the abstract part, the below sentence might mislead the reader about the resolution on account of the working principle of photoacoustic.

“””Hence, photoacoustic imaging has both advantages, i.e.,  high contrast from optical absorption and high spatial resolution from ultra-sound “””.  Actually, the high spatial resolution does not result from ultrasound, it depends basically on optical spot. Maybe, authors wanted to indicate “high spatial resolution, especially in the axial direction” I would suggest to use the terms in optoacoutic field, not the general terms for the manuscript.

  • This is good comment. We wanted to highlight the benefit of combining optics with low scattering ultrasound, which was not clearly written in this sentence. Biological tissues are highly scattering photons, but low scattering ultrasound. Thugs, PAI has higher spatial resolution which is due to lower scattering of acoustic waves. To better clarify this, we changed the sentence as “… and high spatial resolution due to low scattering of ultrasound in tissue”.

As below paragraph from the manuscript, the authors indicated more specific sentences that are need to be support these sentences by the papers, and maybe in the manuscript, the authors already have them, but it should be cited in a particular way after claims, at least end of the paragraph, especially in the reviewer papers.

“Another key component in PAI is the US detector used to receive the PA signal. These 91 detectors can be divided into two categories: piezoelectric transducers and optical sensors. 92 Usually, piezoelectric transducers have a higher sensitivity and are more convenient for 93 an imaging setup, while optical sensors immunize electromagnetic interference and have 94 a wider response bandwidth, but lower sensitivity. Piezoelectric transducers are the most 95 common type of US detector in PAI, due to higher sensitivity, simpler signal detection 96 and sufficient bandwidth. Indeed, bandwidth and the central frequency of the piezoelec- 97 tric transducer determine the axial resolution of PAM. OR-PAM typically employs a 50 - 98 70 MHz central frequency transducer, whereas AR-PAM has a lower frequency, such as 99.”

30 - 50 MHz. For PAMac and PAT, the center frequency of the transducer or array should 100 be as low as 1-10 MHz for detecting lower frequency PA signals from deep tissues with 101 less US attenuation.

  • We added several related references to the sentences there.

*In the part of key advances in PAI, I don’t see any references related fiber laser. In the other parts, even if the manuscript include some references, (such as Allen et al. etc) in the other parts, the references should be also in this part. The properties of excitation source in the field of photoacoustic have a great role on the different applications, in recently, they are fiber excitation source that can be mentioned (Yavas, S, Scientific reports6(1), 1-10). Additionally, as you see in the below sentence, it should be referenced, especially if someone indicate specific values, and studies. Please check whole paper for this problem.

  • in the part of key devices in PAI, we decided to only generally describe the requirements of important parameters for laser sources in PAI, not the laser source categories such as solid, diodes, or fiber laser. In the part “Achievement of PAI techniques in the past five years”, we mentioned the specific laser application in PAI, so we added related references in there. The paper suggested by the reviewer is very good, thus we added this reference in the text in line 230.

“Although the effect of the pulse to-pulse energy stability of PAI can be calibrated by a photodetector, an energy stability better than 2~3% is necessary for a quantitative PA measurement of chromophores in a tissue. “

  • For quantitative PA measurement, the light source should be as stable as possible (2~3% value is our recommendation, not from any specific references). So, we change it in the text as “an energy stability better than 2~3% is recommended for a quantitative real-time measurement of chromophores in a tissue without fluence calibration”

*Although the figure 1 illustrated schemes of PAI techniques with respect to spatial resolution and imaging depth in a feasible way, is there any specific reason why the authors selected the key point of the paper as the study of brain activities, particularly in optoacoustic field? I would suggest the authors to check the important reviews (for example, Karlas, A., Fasoula, N. A., Paul-Yuan, K., Reber, J., Kallmayer, M., Bozhko, D., ... & Ntziachristos, V. (2019). Cardiovascular optoacoustics: From mice to men–A review. Photoacoustics, 14, 19-30) to extend the applications and emphasize the reason why we need to improve or focus on the imaging technology compared to the other technologies for the study of brain injuries, or disease such as DOT or other applications. DOT should be mention in the manuscript, especially brain application and also mentioned some prominent studies based on combined DOT and PAT.

 - This is a good comment. We have added now new a paragraph to introduction, where we emphasize the need to improve imaging technology for the specific applications, mentioned also in the suggested article. Moreover, why especially PA (including DOT) is one of the most potential technologies for this need.

  • So, we change “brain” as “target tissue” in figure 1 for more general description.

Round 2

Reviewer 2 Report

I am still of the opinion that this is way too broad a scope and title for the seemingly disconnected research reviewed in the paper. The "last five" years are not the problem, that is fine. But they just review 3 arbitrary topics: "PAI in small animal brain", "PAI contrast agents" and "combination with other modalities". If they picked one and focused their review on this one sub-field, this wound be fine (though other recent reviews exist for PAI in small animal brain for example). But in this current form these are 3 disconnected, unfinished reviews.

Author Response

  • Thanks again. Based on our knowledge, brain research is one of the hottest fields nowadays, but commonly used techniques such as MRI, CT and ultrasound have some drawbacks, for example, ionized radiation, high cost, or low contrast. PAI has not these problems and is widely used in this field especially in recent years. However, the PAI technique and devices to be used in brain are still suitable for imaging other body parts such as breast, back, or limb. Hence, in the title of this review, we used “recently technical progression in PAI”. We are afraid that the readers misunderstand the mentioned PAI techniques only suitable for brain imaging but not suitable for other imaging, if we use title such as “recently technical progression in PAI in small animal brain”.
  • After introducing PAI categories and the key devices, in order we reviewed the pure (or endogenous, or label-free) PAI in Section 2, contrast enhanced (by exogenous, contrast agents) PAI in Section 3, and combined PAI with other techniques in Section 4. We think that connecting the three parts gives full description of PAI technical progression, although we focused on small animal brain as the examples for avoiding huge PAI application in preclinic and clinic. We hope this review lets the readers who have not yet high expertise/experience in utilising PAI techniques can know the key points of the technical aspects.
  • To accept the reviewer’s suggestion, we limit the review title as “Recent technical progression in photoacoustic imaging - towards using contrast agents and multimodal techniques”.
  • In the end of Abstract section, we modified as “In this review, we focus on advances made in PAI in the last five years and present categories and key devices used in PAI techniques. In particular, we highlight the continuously increasing im-aging depth achieved by PAI, particularly when using exogenous reagents. Finally, we discuss the effects of combining PAI with other imaging techniques.” Please see the resubmitted version which has been now proofread.

Reviewer 3 Report

I thank the authors for replying to all the comments made by me.  I’m happy with some of the corrections to the text.

Author Response

  • Thank you again for the valuable comments for our manuscript.